# *Fusarium*, *Scedosporium* and Other Rare Mold Invasive Infections: Over Twenty-Five-Year Experience of a European Tertiary-Care Center

**DOI:** 10.3390/jof10040289

**Published:** 2024-04-15

**Authors:** Marie-Pierre Ledoux, Elise Dicop, Marcela Sabou, Valérie Letscher-Bru, Vincent Castelain, François Danion, Raoul Herbrecht

**Affiliations:** 1Department of Hematology, Institut de Cancérologie de Strasbourg, 67033 Strasbourg, France; 2Clinics of Oncology, Elsan, 67000 Strasbourg, France; 3Laboratoire de Parasitologie et Mycologie Médicale, Plateau Technique de Microbiologie, Hôpitaux Universitaires de Strasbourg, 67000 Strasbourg, France; 4Institut de Parasitologie et de Pathologie Tropicale, UR 3073 Pathogens-Host-Arthropods-Vectors Interactions, Université de Strasbourg, 67000 Strasbourg, France; 5Intensive Care Unit, Hôpitaux Universitaires de Strasbourg, 67000 Strasbourg, France; 6Department of Infectious Diseases, Hôpitaux Universitaires de Strasbourg, 67000 Strasbourg, France; 7INSERM UMR-S1109, 67000 Strasbourg, France

**Keywords:** invasive mold disease, fungal infection, *Fusarium*, *Scedosporium*, *Alternaria*, *Phaeohyphomycetes*, *Hyalohyphomycetes*, immunocompromised patients

## Abstract

Invasive mold infections (IMD) are an emerging concern due to the growing prevalence of patients at risk, encompassing but not limited to allogeneic hematopoietic stem cell transplant recipients, hematological malignancies patients, solid organ transplant recipients and intensive care unit patients. In contrast with invasive aspergillosis and mucormycosis, other hyalohyphomycoses and phaeohyphomycoses remain poorly known. We conducted a retrospective analysis of the clinical, biological, microbiological and evolutive features of 92 IMD having occurred in patients in our tertiary-care center over more than 25 years. A quarter of these infections were due to multiple molds. Molds involved were *Fusarium* spp. (36.2% of IMD with a single agent, 43.5% of IMD with multiple agents), followed by *Scedosporium* spp. (respectively 14.5% and 26.1%) and *Alternaria* spp. (respectively 13.0% and 8.7%). Mortality at day 84 was higher for *Fusarium* spp., *Scedosporium* spp. or multiple pathogens IMD compared with *Alternaria* or other pathogens (51.7% vs. 17.6%, *p* < 0.05). Mortality at day 84 was also influenced by host factor: higher among hematology and alloHSCT patients than in other patients (30.6% vs. 20.9% at day 42 and 50.0% vs. 27.9% at day 84, *p* = 0.041). Better awareness, understanding and treatments are awaited to improve patient prognosis.

## 1. Introduction

Data and studies about fungal infections among immunocompromised patients have long been focused on invasive aspergillosis [1]. More recently, the emergence of mucormycosis [2] has led to an increasing awareness about non-*Aspergillus* mold infections. As a step forward into better understanding rare pathogen infections, we decided to investigate the cases of rare-mold invasive infections that have occurred in patients at our institution. Concerns about their increasing burden have actually been expressed in recent literature, either in epidemiological studies [3,4,5] or in international guidelines [6,7,8]. Fungi involved in these infections are numerous, and include *Phaeohyphomycetes* such as *Alternaria*, *Cladophialophora*, *Cladosporium*, *Phaeoacremonium*, *Lomentospora* and *Hyalohyphomycetes* such as *Fusarium*, *Scedosporium*, *Scopulariopsis*, *Penicillium*, *Paecilomyces* or *Trichoderma* [9].

On the other side of this undesired encounter are usually found severely immunocompromised patients [10], mostly from hematological background with or without allogeneic hematopoietic stem cell transplant (alloHSCT) [11,12,13,14] or solid-organ-transplant (SOT) recipients [15,16,17]. Newer frames of immunodepression are being described [18,19,20,21,22] and are expected to increase the number of patients at risk for invasive mold infections (IMD). Moreover, changes in the antifungal strategies applied to these patients, for instance prophylaxis, might have a strong impact on the epidemiology of breakthrough infections [23].

Data are scarce about diagnostics, management and outcome of these severe conditions. Several studies have however enhanced our knowledge by focusing on one pathogen, such as *Fusarium* [3,24,25], *Scedosporium* [4,26] or *Alternaria* [5]. Throughout this literature, we are faced with the fact that prognosis remains poor for these rare infections, and more data is awaited to better diagnose and treat the patients concerned. With this article, we try to bring in our contribution through the description of *Fusarium*, *Scedosporium* and other rare mold invasive infections having occurred in our tertiary-care center from 1997 to 2023.

## 2. Patients and Methods

We conducted a retrospective analysis based on clinical reports, biological and microbiological data and radiological results of all patients diagnosed with probable or proven invasive mold disease, abiding by EORTC-MSGERC criteria [27], excluding *Aspergillus* and Mucorales infections. We did not retain possible IMD due to the impossibility of ruling out *Aspergillus* spp. or Mucorales as pathogens in these cases. Colonization with rare molds were also not retained. Cases were identified through requests to Medical Information Systems Programs (MISP) and to the information system of the mycological laboratory. Data regarding demographic features, underlying conditions and medication, diagnostic features, microbiological analysis, treatment and outcome were gathered, as part of the study of opportunistic infections approved by the institutional ethics committee of the Hôpitaux Universitaires de Strasbourg (FC2017-17). According to French regulations, the database was declared to the Commission Nationale de l’Informatique et des Libertés. The study was registered at ClinicalTrials.gov as no. NCT03920735.

Mycological diagnosis of rare fungal infections was performed using a combination of direct examination, fungal culture and antigen detection in various samples including blood, broncho-alveolar lavage fluid (BALF), or biopsy materials when available. Identification of the fungal pathogens was obtained using morphology studies and sequencing technics of relevant genes comprising elongation factor EF1alpha, ribosomal DNA ITS region, or calmodulin. Serum and BALF galactomannan was assessed with Platelia Bio-Rad kit (Bio-Rad, Marnes-la-Coquette, France). Serum beta-D-glucan was assessed with Fungitell kit (Associates of Cape Cod Inc., East Falmouth, MA, USA). Antifungal sensitivity was assessed through the determination of minimal inhibitory concentration (MIC) with a gradient method.

Although some fungi have changed name in the nomenclature, we chose to keep the names given on diagnosis to avoid splitting categories that make sense from a clinical perspective, all the more so when fungi were identified only to the genus level. A correspondence chart is shown in Table 1.

Statistical analyses were performed with GraphPad Prism 10.1.1. (GraphPad Software, Boston, MA, USA). Characteristics of the patients were compared using frequencies, medians and means. For the comparison analysis, Chi-square, Fisher’s exact test or *t*-test were used as appropriate. Kaplan-Meier method and log rank test were used to estimate and compare survivals. The *p*-value was considered statistically significant if <0.05.

## 3. Results

We identified 92 cases of probable or proven rare mold infections among 89 patients, taken care of in *Hôpitaux Universitaires de Strasbourg*, France, between January 1997 and December 2023. Demographic features and observed risk factors are detailed in Table 2. All but one of the 21 alloHSCT patients were transplanted for a hematological malignancy. Thirteen patients (61.9% of alloHSCT patients) presented with graft vs. host disease (GVHD) at the time of the IMD (of which 46.2% acute and 53.8% chronic GVHD). Most frequent hematological malignancy among the hematological patients without alloHSCT was acute myeloid leukemia (11 patients—39.3%), followed by aggressive lymphoma (seven patients—25%, of which two had recently undergone autologous stem cell transplant). Among 24 SOT recipients, organ transplanted was kidney for nine patients (37.5%), liver for eight patients (33.3%), lung for seven patients (29.2%) and heart for two patients (8.3%). One patient was transplanted with both kidney and liver, and one patient was transplanted with both kidney and heart. Benign hematological or immunological disorders were aplastic anemia, toxic agranulocytosis, chronic granulomatous disease, rheumatoid arthritis and polymyalgia rheumatica. Other main host factors were prosthetic aorta, pulmonary surgery, chronic obstructive pulmonary disorder, hepatic failure, sepsis or mold exposure through gardening, walking barefoot or near drowning in a pond.

### 3.1. Fungi and Clinical Presentation

Fungi identified as pathogens in our cohort are shown in Figure 1. Fungi involved in IMD were of one species in 75% of cases and two or more species in 25% cases. Details are provided in Table 3. Most prevalent fungus among our cohort was *Fusarium* spp. (36.2% of IMD with a single agent, 43.5% of IMD with multiple agents), followed by *Scedosporium* spp. (respectively 14.5% and 26.1%) and *Alternaria* spp. (respectively 13.0% and 8.7%). *Paecilomyces* spp. was only found in multiple-mold IMD.

IMD were proven in 53 (57.6%) cases and probable in 39 (42.4%) cases, as per EORTC/MSG criteria. Lung involvement was found in 56 cases (60.9%). It was more frequent in hematology patients and alloHSCT patients than in other patients (81.0%, 75.0% and 44.2% respectively, *p* = 0.004), and more frequent without reaching statistical significance in lung transplant patients compared with other SOT patients (71.4% vs. 35.3%, *p* = 0.104). Main fungi involved in lung IMD were *Fusarium* spp. (50.0%), *Scedosporium* spp. (23.2%) and *Alternaria* spp. (8.9%). Acral localization (i.e., with supposed inoculation in limbs, limited to skin or not, with secondary dissemination or not) was seen in 15 (16.3%) patients, mostly SOT patients (60.0%). Main fungus involved in acral localization was *Fusarium* spp. (33.3%). Blood stream infections were documented in 10 patients (11.0%). Fungus involved was *Fusarium* spp. in 80.0% of fungemia. The remaining two fungemia-involved fungi were *Trichoderma longibrachiatum* and *Phaeoacremonium*. *Fusarium* spp. was also the most prevalent fungus involved in disseminated infections (46.7%), followed by *Scedosporium* spp. (23.3%), *Penicillium* spp. and *Alternaria* spp. (10.0% each). A case of *S. apiospermum* disseminated IMD is described in Figure 2.

### 3.2. Biomarkers

Galactomannan assay in serum (GMs) was available in the diagnostic panel of 72 IMD (78.2%), and positive in 22.2% of these. Among the positive cases, 62.5% were IMD involving culture-identified *Aspergillus* spp. as well as a rare mold. The remaining cases were two IMD with *Alternaria* spp. (among 5 *Alternaria* IMD with available GMs), two with *Fusarium* spp. (among 29 *Fusarium* IMD with available GMs), one *Geosmithia argillacea* and one *Penicillium* sp. (among 5 *Penicillium* IMD with available GMs). Of note, none of the 7 *Scedosporium* IMD for which galactomannan in serum was available showed a positive result.

Galactomannan assay in BALF was available in the diagnostic panel of 26 IMD (28.2%), and positive for 34.6% of these. Among the positive cases, 66.7% were IMD involving *Aspergillus* spp. as well as a rare mold. The remaining cases were an IMD with a dematious mold not further identified, the afore mentioned IMD with *Geosmithia argillacea* and a case of pulmonary co-infection by *Rhizopus oryzae*, *Fusarium proliferatum* and *Scedosporium apiospermum*.

Beta-D-glucan assay was available from 2019 in our center, and part of the diagnostic panel of 10 IMD (11.0%), and positive for 50.0% of these: an IMD with *Alternaria alternata*, an IMD with *Fusarium verticillioides*, two IMD with *Scedosporium apiospermum* and an IMD with *Penicillium chrysogenum*. Conversely, beta-D-glucan assay was positive for all of the *Alternaria* IMD, a third of the *Fusarium* IMD, half of the *Penicillium* IMD and all of the *Scedosporium* IMD in which it was available. 

### 3.3. Antifungal Susceptibility

Antifungal susceptibility testing was available for 52 strains obtained from culture. MIC were higher than 32 mg/L in 24/52 species (46%) and higher than 2 mg/L in 36/52 species (69%) for amphotericin B. MIC were higher than 32 mg/L in 14/52 species (27%) and higher than 2 mg/L in 25/52 species (48%) for voriconazole. MIC were higher than 32 mg/L in 22/52 species (42%) and higher than 2 mg/L in 31/52 species (60%) for posaconazole. MIC were higher than 32 mg/L in 31/52 species (60%) and higher than 2 mg/L in 34/52 species (65%) for caspofungin. More detailed data are shown in Table 4.

### 3.4. Treatment

Median treatment delay after the first signs of infection in the whole cohort was 7 days (range 0–325 days; interquartile range IQR 3–16 days). It was significantly longer in SOT patients than in hematology and alloHSCT patients: 14 days vs. 3 days (*p* = 0.0006). Treatment comprised an antifungal in 95.7%, the remaining patients having been treated by surgery only (2 patients) or not treated due to diagnosis being made post-mortem (2 patients). The most frequently used antifungal as a single agent was voriconazole (47.9%) followed by amphotericin B deoxycholate or lipidic formulation (30.1%—respectively 5.5% and 24.6%), caspofungin and posaconazole (6.8% each). Combination therapy was started as a first treatment in 16.3% of IMD and comprised mostly voriconazole and caspofungin (33.3%). A change of treatment within the first 30 days was observed in 41 cases (46.6%). Among them, a change of treatment towards a combination therapy was seen in 17 cases (41.5%). The median duration of treatment was 48 days (range 1 to 1078). Treatment was stopped after clinical success in 37 cases (40.2%). Surgery was part of the treatment in 23 cases (25%), of which 39.1% for acral localization, 17.4% for sinusal localization and 17.4% for pulmonary localization. Conversely, lung surgery was done in 1 of 39 (2.6%) IMD involving lung only and 2 of 17 (11.7%) IMD involving lung and other sites. Sinus surgery was done in 4 of 5 (80.0%) IMD involving sinus only and 2 of 7 (28.6%) IMD involving sinus and other sites. Local surgery was done for 8 of 12 (60.0%) acral localization of IMD without dissemination and 1 of 3 (60%) acral localization of IMD with secondary dissemination. It was for instance part of the treatment of the acral localization of *Fusarium solani* described in Figure 3. Overall survival at day 84 is 83.3% for patients having undergone surgery, compared with 52.9% for patients who have not (*p* = 0.009).

### 3.5. Outcome

Median follow-up was 226 days, i.e., 7.4 months, with a range from 4 days to 22.6 years and an IQR from 42 days to 2.6 years. At the end of the follow-up period, 73.9% of patients had died, with 38.2% of deaths attributable to the IMD itself. Thirty-four patients (50.0%) were considered to have died from another cause with no sign of active IMD. Among 24 deaths occurring up to day 42, 23 are attributable to IMD (95.8%) and 1 to another cause, whereas among 12 occurring between day 43 and day 84, 8 are attributable to IMD (66.7%) and 4 to other causes (*p* = 0.017).

Day 42 mortality was 26.1%, and day 84 mortality was 39.1%. Survival estimates for the whole cohort within the first 12 weeks and over the whole follow-up are shown in Figure 4a,b upper panel.

Mortality was higher among hematology and alloHSCT patients than in other patients (30.6% vs. 20.9% at day 42 and 50.0% vs. 27.9% at day 84, *p* = 0.041). Survival estimates comparing the outcome according to main host factor at week 12 and over the whole follow-up are shown in Figure 4c,d second panel from top.

Mortality at day 42 was 24.0% for *Fusarium* spp., 30.0% for *Scedosporium* spp., 22.2% for *Alternaria* spp., 8.0% for others and 47.8% for IMD with several pathogens. Mortality at day 84 was 44.0% for *Fusarium* spp., 50.0% for *Scedosporium* spp., 22.2% for *Alternaria* spp., 16.0% for others and 60.9% for IMD with several pathogens. There was a significant difference between mortality of IMD by *Fusarium* spp., *Scedosporium* spp. and several pathogens on one hand and *Alternaria* spp. and other pathogens on the other hand: mortality was 34.5% at day 42 and 51.7% at day 84 for the first group and respectively 11.8% and 17.6% for the second group (*p* = 0.0022). Survival estimates comparing the outcome according to the fungi involved at week 12 and over the whole follow-up are shown in Figure 4e,f third panel from top.

Mortality at day 42 was 20.5% for pulmonary localization only, 25.0% for other focal IMD, 16.7% for disseminated excluding lung and 47.1% for disseminated including lung. Mortality at day 84 was 38.5% for pulmonary localization only, 25.0% for other focal IMD, 41.7% for disseminated excluding lung and 58.8% for disseminated including lung. Survival estimates comparing the outcome according to localization at week 12 and over the whole period are shown in Figure 4g,h lower panel.

### 3.6. Trends over Time

Occurence of rare mold invasive infections is increasing over time in our cohort. Among 92 episodes of IMD, 18 (20%) occurred between 1997 and 2006, 38 (41%) occurred between 2007 and 2016 and 36 (39%) occurred in the shorter period between 2017 and 2023. The increase is stronger for *Fusarium* sp. with occurrences of respectively 3, 11 and 11 infections involving *Fusarium* sp. only in these time periods. Occurrences of infections involving *Scedosporium* sp. only also show an increase with respectively 2, 3 and 5 episodes over the said time spans. More detailed data are available in Figure 5. Comparison of survival estimates by Kaplan Meier method did not show a significant difference between patients diagnosed between 1997 and 2010 and patients diagnosed after 2011.

## 4. Discussion

Although worrisome in terms of prognosis, IMD involving *Fusarium* spp., *Scedosporium* spp. and other rare molds are rare events. To our knowledge, our cohort describing 92 IMD having occurred over 25 years to patients with diverse underlying diseases is the widest of the kind to be published so far. 

Overall survival reaches 73.9% at week 6 and 60.9% at week 12, which compares favorably with the literature, with a usual 50% of overall survival [24,25]. This is probably a reflection of the heterogeneity of the patients in our cohort. The diversity of host factors exhibited in our cohort enables us to assess differences between the various patient populations. Survival is indeed heterogenous, significantly better for SOT patients and other patients than hematology and alloHSCT patients (mortality being 28% vs. 50% at day 84, *p* < 0.05). Attributability of death analysis shows that the proportion of deaths attributable to IMD is significantly higher within the first 6 weeks compared to the 6 following weeks (95.8% vs. 66.7%, *p* < 0.05). However, IMD is still the most prevalent cause of deaths occurring between day 43 and day 83, which is a higher proportion than what has been described for invasive aspergillosis [28]. Importantly, our data shows how causes of late death are often other factors than IMD itself, which highlights that prognosis is a complex issue: however curable, IMD is prone to interfere with the management of the underlying disease and can lead to harmful delays.

It also appears that diagnosis and treatment of IMD are more of an emergency in hematology and alloHSCT patients than in SOT patients, who have a significantly longer delay from first signs to treatment (median delay 14 days vs. 3 days, *p* < 0.05) and nevertheless a better outcome, as had already been described in the TRANSNET analysis [17]. This can point towards a difference in natural history of the IMD, being more symptomatic and rapidly severe in hematology and alloHSCT patients. One can hypothesize the mechanism underlying has to do with neutropenia as a risk factor [10]. Our cohort is actually in contrast with many others in terms of neutropenia frequency that reaches only 28.3%, compared with 75% in a 233 fusariosis cohort described by Nucci et al. [29] or 58.8% in an alternariosis cohort described by Pastor et al. [30]. Interestingly, our patients presented with a high prevalence of lymphopenia (67.4%) which is not yet clearly reckoned as a risk factor for IMD, although recent update of EORTC/MSG criteria for definition of host factors [27] tends to encompass T-cell and B-cell deficiency in the immune disorders leading to a rise in IMD risk.

The usefulness of galactomannan in making a differential diagnosis between aspergillosis and other IMD is controversial [31]. On one hand, sensitivity of serum galactomannan has dropped in the hematological population since prophylaxis has come to a wide use in neutropenic patients. On the other hand, galactomannan can’t be considered specific for *Aspergillus* spp., and is sometimes regarded as a criterion for *Fusarium* spp. infection [32,33], although others rather considered its lack of positivity as a criterion [3], to finally conclude that positive galactomannan should be interpreted cautiously [34]. In our cohort, galactomannan was positive in serum or BALF for some, yet not all infections by *Alternaria* spp., *Fusarium* spp., *Geosmithia argillacea*, *Penicillium* spp. and for a case of pulmonary co-infection by *Rhizopus oryzae*, *F. proliferatum* and *S. apiospermum*, showing how diverse the final diagnosis can be when confronted with a positive galactomannan. More frequently, it was positive in cases of co-infections with *Aspergillus* spp., which should be a subject of particular vigilance [35]. An unexpected positive galactomannan could also be the sign of a yet unretrieved *Aspergillus*. Data are very limited in our cohort as far as beta-D-glucan is concerned, due to a rather recent availability in our center. It was positive for an IMD with *A. alternata*, an IMD with *F. verticillioides*, two IMD with *S. apiospermum* and an IMD with *P. chrysogenum*, only consistent with its well-known non-specificity.

*Fusarium* spp. was the most frequent fungus involved in pulmonary IMD, acral IMD, fungemia and disseminated infections in our cohort. It is indeed, after *Aspergillus* spp. and Mucorales, the third mold responsible for IMD in many studies [14,16,23] and can also be involved in superficial mycoses with a high prevalence [36]. Among 25 IMD involving *Fusarium* spp. only, 7 led to positive blood cultures. An 8th fungemia was observed in a patient with concomitant pulmonary aspergillosis. This high rate of blood culture positivity is a characteristic of *Fusarium* spp. [34] and should lead to ensure a sufficient number of blood tests when diagnosis is suspected, all the more so when the responsible fungus is yet to be determined. Moreover, a prolonged culture is necessary to improve the yield of blood cultures: seven days are required. Biopsy of skin lesions is another high-yield-associated diagnostic procedure. Survival was 76% at day 42 and 56% at day 84, which is consistent with data from the recent literature [24,25,37].

Second in terms of prevalence in our cohort, *Scedosporium* spp. comes in the first place in terms of mortality, ranking 30% at day 42 and 50% at day 84, only topped by multiple fungi infections (in which it often takes part). These results are close to those already described in analyses of FungiScope in both pediatric [38] and adult [4] population: respectively 31% at day 42 and 22.2% then 43.6% at day 42 and 84. This poor outcome is a result of a strong proneness to disseminated infections and a usual pan-antifungal resistance. A specific trend to cerebral and cardiac involvement also participates in usually dismal prognosis, as shown in a recent cohort [39]. Disseminated infections are usually associated with a high rate of fungemia [40], which was not observed in our cohort. Association with other pathogens has also been previously described [41]. Recently pointed out of the *Scedosporium* genera, *L. prolificans* is of particularly adverse prognosis in literature [26]. Analysis of our cohort through this prism is limited, since only one IMD led to the precise diagnosis of *L. prolificans*, within a co-infection with *S. apiospermum* and *Lichtheimia* sp. and was lethal. *Scedosporium spp.* we met were mostly *S. apiospermum*, with the exception of one *S. dehoogii* and one *S. aurantiacum*.

*Alternaria* spp. infections have previously been associated with SOT, hematological malignancy and alloHSCT [5,30,42]. More than a third of the patients diagnosed with alternariosis in our cohort had however none of these host factors: among eleven patients, one had aplastic anemia, one had toxic agranulocytosis, one had severe hepatic failure and the fourth had heavily treated rheumatoid arthritis. Relatively better outcomes usually described in literature is observed in our cohort, with a 42-day survival of 77.8% and a 84-day survival stable at 77.8%. Other rarer pathogens also show a relatively favorable outcome with a 42-day survival of 92% and a 84-day survival of 84%. These findings seem consistent with the description of extremely rare invasive fungal infections reported in FungiScope [43]. It is in contrast with a rather poor outcome described in literature for *Phaeooacremonium* sp. [44] or *G. argillacea* [45].

Of particularly poor prognosis are infections by several molds, with a mortality ranking 47.8% at day 42 and 60.9% at day 84. Although those infections are numerous in our cohort, representing 25% of the IMD described, data is still insufficient to understand whether this poor prognosis is due to one species in particular (such as *Scedosporium* spp. or *Fusarium* spp., often observed in co-infections) or rather to an additive effect of the fungi. This high incidence of co-infections should, all the more so when associated with poor prognosis, lead to a particular attention on diagnostic work-up: finding proof for one fungus does not rule out other fungi, and an unexpected unfavorable evolution should prompt further examinations, not to disregard a complex diagnosis [35]. Of note, the fact that all the *Paecylomyces* sp. infections in our cohort are multiple fungi infections does not seem to be a previously reported trend of this fungus [46].

Localization of the IMD also plays an expected role in the prognosis: disseminated IMD, all the more so when it includes a pulmonary localization, is associated with a poor outcome.

Data about MIC and antifungal susceptibility of rare molds are scarce in the literature. Our cohort brings some noteworthy trends that must not be over interpreted. Clinical breakpoints are often yet undetermined for rare molds, further complexifying the reading of MIC. *Fusarium* spp. is associated in our cohort with an expected dreadful susceptibility profile, as well as *Scopulariopsis* spp. Voriconazole seems to be an interesting option for the treatment of *Scedosporium* spp. as previously described [47]. As it has only been recently made available, isavuconazole has only been tested against a few strains in our cohort. Among the 25 fungi tested, isavuconazole does not appear more potent than voriconazole, all the less so for *Scedosporium* spp. or *Trichoderma* spp. For the latter, caspofungin and voriconazole exhibit interesting activity, as shown in literature [48]. Amphotericin B met no resisting mold among the few strains of *Alternaria* spp., *Chrysosporium* sp. and *Penicillium* spp. we were given to test. In the four cases for which it was added to the tests, flucytosine did not come out useful. A proportion of susceptibility testing could not be done due to the invasion of cultures by other species involved the same infection, underlying once again the challenges of co-infections.

The most frequently used antifungal as a single agent in our cohort was voriconazole, followed by amphotericin B, whose galenic formulations varies over time in our long-period observational cohort. The high rate of combination therapies (16.3%) and early switches of treatment (46.6%) makes it impossible to draw a conclusion on the efficiency of the various antifungal used. Combination therapy seems a promising option in rare mold IMD [49] but needs further investigation in a frame where randomized clinical trials are a challenge. Surgery should not be neglected, as it has been associated with better outcomes in several cohorts [26,36] as well as in ours, but its feasibility is often by itself a marker of less severe IMD, which induces a bias in analysis.

As our cohort extends over a long-time span, we expected to be able to see some trends over time. In terms of occurrence, it is obvious that rare molds infections increase along years. It is even more so for *Fusarium* sp. and *Scedosporium* sp., which is congruent with the fact that both of them are often being called emergent pathogens. However, our study could not bring data to assess the proper incidence of these IMD, by the lack of a denominator: this apparent increase could also be related to better diagnostic tools or to better awareness over time. Of note, we did not observe an increase in cases reported in 2020, contrasting with what has been reported world-wide for aspergillosis and mucormycosis through COVID-19-associated mold infections [50].

In terms of outcome, there was no significant difference between the first and the second halves of our cohort. A pessimistic interpretation of this could be that no improvement has been done, but some bias has to be taken into account: some missed diagnosis cases of the earlier part of the cohort could precisely be poor outcome patients for whom no autopsy was performed, and a larger part of the more recent patients bear poor-outcome associated host factors such as deeper and longer immunosuppression. Nevertheless, there is no doubt that some improvement can and must be done.

## 5. Conclusions

As more and more patients present with risk factors such as hematological malignancy, alloHSCT, SOT or newer forms of immunodepression, and as less and less aspergillosis fail to answer favorably to prophylaxis and treatment, clinicians are bound to meet more and more *Fusarium* spp., *Scedosporium* spp. and other so-called rare molds. Although their presentation can mimic aspergillosis or mucormycosis, their clinical features tend to be more diverse, and their management, from diagnostic work-up to treatment, needs particular attention. Our study, by showing diversity of both patients concerned and molds involved, intends to raise awareness and take part in improving knowledge of rare invasive mold diseases, to eventually improve prognosis.

## Figures and Tables

**Figure 1 jof-10-00289-f001:**
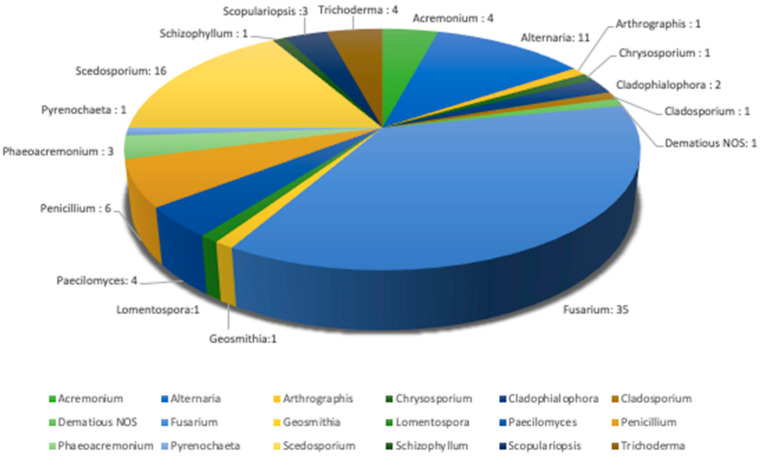
Pie chart showing the distribution of the species of fungi identified. Among them, *Acremonium* were 1 *A. brunnescens*, 1 *A. strictum*, 2 *Acremonium* sp.; *Alternaria* were 4 *A. alternata*, 1 *A. infectoria*, 6 *Alternaria* sp.; *Arthrographis* was *A. kalrae*; *Cladophialophora* were 2 *C. bantiana*; *Fusarium* were 2 *F. dimerum*, 7 *F. fujikuroi*, 2 *F. moniliforme*, 1 *F. oxysporum*, 1 *F. petroliphilum*, 5 *F. proliferatum*, 4 *F. solani*, 1 *F. verticillioides* and 12 *Fusarium* sp.; *Geosmithia* was *G. argillacea*; *Lomentospora* was *L. prolificans; Paecilomyces* were 1 *P. variotii* and 3 *Paecilomyces* sp.; *Penicillium* were 1 *P. citrinum*, 2 *P. chrysogenum* and 3 *Penicillium* sp; *Phaeoacremonium* were 1 *P. fuscum* and 2 *Phaeoacremonium* sp; *Pyrenochaeta* was *P. romeroi*; *Scedosporium* were 1 *S. aurantiacum*, 14 *S. apiospermum*, and 1 *S. dehoogii*; *Schizophyllum* was *S. commune*; *Scopulariopsis* were 1 *S. brevicaulis*, 1 *S. gracilis* and 1 *Scopulariopsis* sp; *Trichoderma* were 3 *T. longibrachiatum* and 1 *Trichoderma* sp.

**Figure 2 jof-10-00289-f002:**
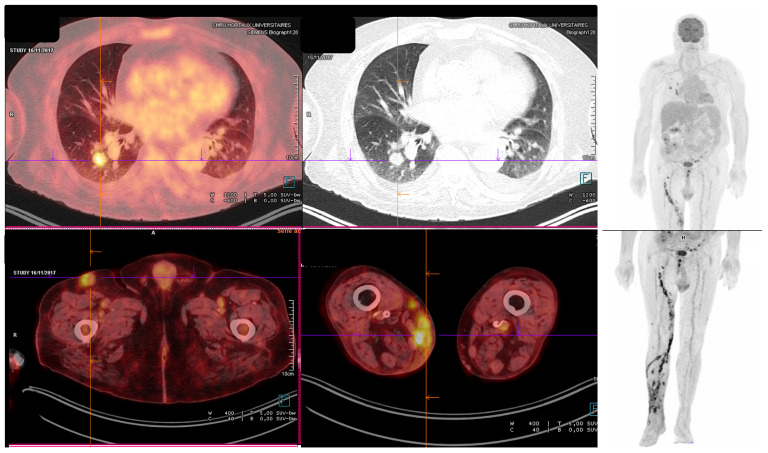
*Scedosporium apiospermum* disseminated IMD occurring in a 80-year-old kidney transplant recipient. Patient presented with a thigh abscess end of October, in which *S. apiospermum* was identified by biopsy at the beginning of November. PET-CT was performed on 16 November 2017 to search for extension and showed whole limb extension and pulmonary localization. BAL could not be performed due to the frailty of the patient. IMD did not improve on voriconazole and patient died on 29 November 2017.

**Figure 3 jof-10-00289-f003:**
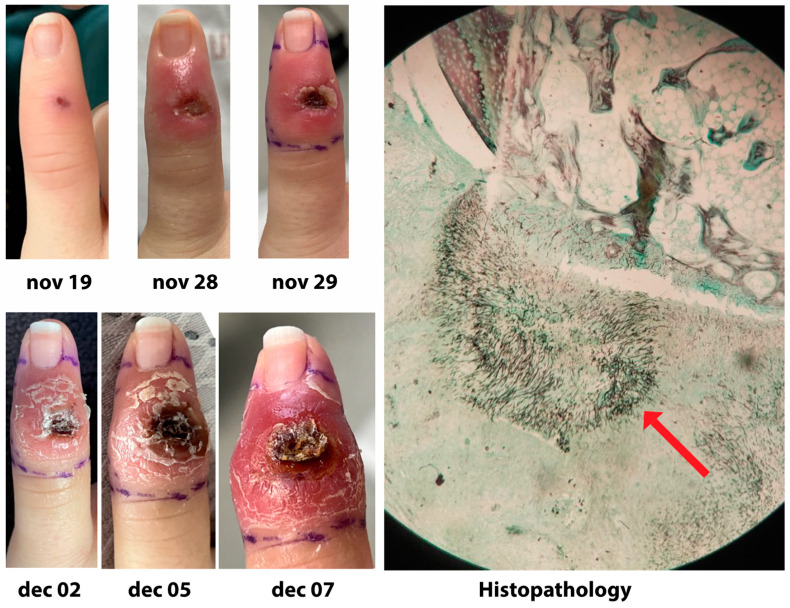
*Fusarium solani* acral invasive mold disease in a 20-year-old woman presenting with refractory AML, and neutropenia for one month. Photographs show a rapid local progression after a minor wound, in spite of the initiation of liposomal amphotericin B and terbinafine on 30 November 2022, after results of pus culture showing *F. solani*. After a switch to voriconazole and terbinafine on 7 December 2022, decision was made to amputate the finger, and histopathology Gomori methenamine silver staining shows numerous hyphae (red arrow). Finger culture grew *F. solani*. Blood cultures remained sterile and patient did well after amputation.

**Figure 4 jof-10-00289-f004:**
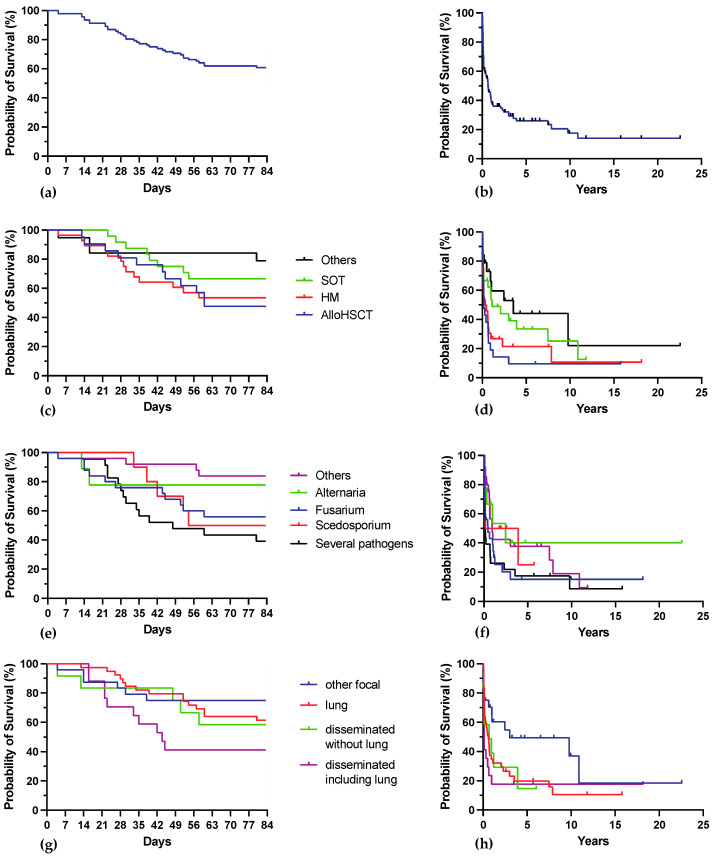
Kaplan-Meier estimates for survival. (**a**) shows 84-day survival for the whole cohort. (**b**) shows overall survival for the whole cohort. (**c**) shows 84-day survival according to main host factor. (**d**) shows overall survival according to main host factor. (**e**) shows 84-day survival according to the identified mold. (**f**) shows overall survival according to the identified mold. (**g**) shows 84-day survival according to IMD localization. (**h**) shows overall survival according to IMD localization.

**Figure 5 jof-10-00289-f005:**
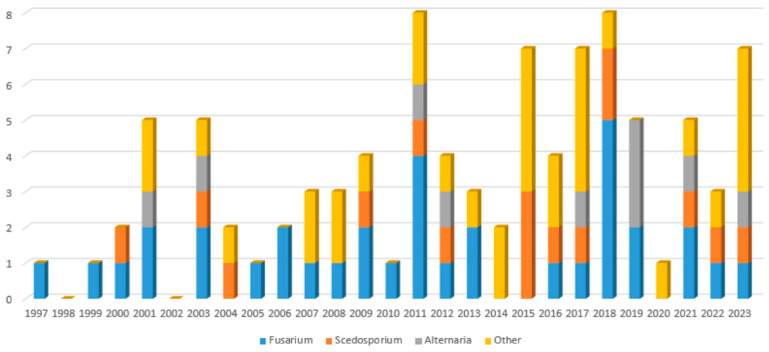
Histogram distribution of the cohort according to year of diagnosis and species involved. Multiple fungi infections are comprised in the category Other.

**Table 1 jof-10-00289-t001:** Correspondence between names used in our study and current nomenclature.

Name Used in the Study	Current Nomenclature
*Acremonium brunnescens*	*Brunneomyces brunnescens*
*Acremonium strictum*	*Sarocladium strictum*
*Fusarium dimerum*	*Bisifusarium dimerum*
*Fusarium petroliphilum*	*Neocosmospora petroliphila*
*Fusarium solani*	*Neocosmospora solani*
*Geosmithia argillacea*	*Rasamsonia argillacea*
*Pyrenochaeta romeroi*	*Medicopsis romeroi*
*Scopulariopsis gracilis*	*Microascus gracilis*

**Table 2 jof-10-00289-t002:** Demographics and risk factors.

Characteristics						N (Range)	%
Median age, years						56 (8; 80)	
Age IQR, years						45–65	
Male/Female						57/35	*62.0*/*38.0*
**Main host factor (N = 92)**							
	Hematological malignancies (without alloHSCT)					28	*30.4*
	Allogeneic hematopoietic stem cell transplant					21	*22.8*
	Solid organ transplant					24	*26.1*
	Benign hematological/immunological disorder					5	*5.4*
	Solid organ cancer					2	*2.2*
	AIDS					1	*1.1*
	Other					11	*12.0*
**Other risk factors**							
	Neutropenia on day of first symptoms (<0.5 G/L)					26	*28.3*
	Median duration of neutropenia before diagnosis (days)					28 (1; 274)	
	Duration of neutropenia IQR (days)					9–45	
	Lymphopenia on day of first symptoms (<1 G/L)					62	*67.4*
	Treatment within the last 90 days before diagnosis						
		Immuno suppressant	Cancer chemo-therapy	Cancer targeted therapy	Cortico-steroids		
		-	-	-	-	14	*15.2*
		+	-	-	-	5	*5.4*
		-	+	-	-	16	*17.4*
		-	-	+	-	5	*5.4*
		-	-	-	+	4	*4.3*
		+	+	-	-	5	*5.4*
		-	+	-	+	6	*6.5*
		+	-	-	+	29	*31.5*
		-	-	+	+	1	*1.1*
		+	+	-	+	7	*7.6*
	Total	46	34	6	47	92	*100*
	Smoking or history of smoking					37	*40.2*
	Prior respiratory diseases					28	*30.4*
	Diabetes mellitus					24	*26.1*
	Mechanical ventilation on day of first symptoms					12	*13.0*
**Antifungal (as prophylaxis or preceding treatment) before diagnosis**							
	Caspofungin					7	*7.6*
	Fluconazole					10	*10.9*
	Itraconazole					2	*2.2*
	Voriconazole *					11	*12.0*
	Posaconazole					12	*13.0*
	Isavuconazole					1	*1.1*
	Amphotericin B (oral)					1	*1.1*
	Amphotericin B (IV) *					2	*2.2*

* One patient had both liposomal amphotericin B and voriconazole before diagnosis. Immunosuppressants are: alemtuzumab, anti-thymocyte globulin, azathioprine, basiliximab, ciclosporine, eculizumab, everolimus, methotrexate (depending on indication), mycophenolate, ruxolitinib (depending on indication), sirolimus, tacrolimus. Cancer chemotherapy are: amsacrine, asparaginase, bendamustin, brentuximab, busulfan, carboplatin, carmustine, chlorambucil, cisplatin, clofarabine, cyclophosphamide, cytarabine, daunorubicin, doxorubicin, etoposide, fludarabine, 5-fluorouracil, gemcitabine, homoharringtonin, hydroxycarbamide, idarubicin, ifosfamide, irinotecan, lomustin, 6-mercaptopurin, methotrexate (depending on indication), mitoxantron, oxaliplatin, rituximab, thiotepa, trabectedin, vincristine. Cancer targeted therapy are: azacitidine, decitabine, midostaurin, ruxolitinib (depending on indication), venetoclax.

**Table 3 jof-10-00289-t003:** Invasive mold disease characteristics.

Characteristics		*N*	%
**Species involved in infections by one mold (*n* = 69)**			
	*Fusarium* spp.	25	27.2
	*Scedosporium* spp.	10	10.9
	*Alternaria* spp.	9	9.8
	*Penicillium* spp.	6	6.5
	*Trichoderma* spp.	4	4.3
	*Phaeoacremonium* spp.	3	3.2
	*Acremonium* spp.	2	2.2
	*Cladophialophora bantiana*	2	2.2
	*Scopulariopsis* spp.	2	2.2
	*Cladosporium* spp.	1	1.1
	*Chrysosporium* spp.	1	1.1
	*Geosmithia argillacea*	1	1.1
	*Pyrenochaeta romeroi*	1	1.1
	*Schizophyllum commune*	1	1.1
	Dematious mold not further identified	1	1.1
**Species involved in infections by several molds (*n* = 23)**			
	*Fusarium* spp. + *Aspergillus* spp.	6	6.5
	*Scedosporium* spp. + *Aspergillus* spp.	4	4.3
	*Acremonium* spp. + *Aspergillus* spp.	2	2.2
	*Paecilomyces* spp. + *Aspergillus* spp.	2	2.2
	*Alternaria* sp. + *Aspergillus fumigatus*	1	1.1
	*Alternaria alternata* + *Fusarium* sp. + *Aspergillus fumigatus*	1	1.1
	*Arthrographis kalrae* + *Aspergillus fumigatus*	1	1.1
	*Fusarium dimerum* + *Lichtheimia corymbifera*	1	1.1
	*Fusarium* sp. + *Paecilomyces* sp.	1	1.1
	*Fusarium proliferatum* + *Scedosporium apiospermum* + *Rhizopus oryzae*	1	1.1
	*Lomentospora prolificans* + *Scedosporium apiospermum* + *Lichtheimia* sp.	1	1.1
	*Paecilomyces* sp. + *Mucor* sp.	1	1.1
	*Scopulariopsis* sp. + *Aspergillus lentulus* + *Rhizomucor* sp.	1	1.1
**Infection site**			
	Pulmonary infection without further extension	39	42.4
	Extra-pulmonary focal infection (Of which 12 acral, 5 sinusal, 5 abdominal, 1 cerebral and 1 tonsillar)	24	26.1
	Disseminated infection without pulmonary involvement	12	13.0
	Disseminated infection with pulmonary involvement	17	18.5

**Table 4 jof-10-00289-t004:** Antifungal susceptibility results.

	Proportion MIC Test Available	Percentage with MIC > 2 mg/L among Available Ones (*Number Tested*)						
		AmB	ITZ	VRZ	PSZ	ISZ	CAS	5FC
*Acremonium* spp.	1/4	100 *(1)*	0 *(1)*	0 *(1)*	0 *(1)*	- *(0)*	100 *(1)*	- *(0)*
*Alternaria* spp.	3/11	0 *(3)*	33 *(3)*	67 *(3)*	0 *(3)*	67 *(3)*	67 *(3)*	- *(0)*
*Chrysosporium*	1/1	0 *(1)*	100 *(1)*	0 *(1)*	0 *(1)*	- *(0)*	0 *(1)*	- *(0)*
*Fusarium* spp.	21/35	86 *(21)*	100 *(21)*	81 *(21)*	90 ^a^ *(21)*	100 *(10)*	100 *(21)*	100 *(2)*
*G. argilacea*	1/1	100 *(1)*	100 *(1)*	100 *(1)*	100 *(1)*	- *(0)*	0 *(1)*	- *(0)*
*Paecylomyces* spp.	2/4	50 *(2)*	50 *(2)*	100 *(2)*	50 ^b^ *(2)*	- *(0)*	50 *(2)*	- *(0)*
*Penicillium* spp.	4/6	0 *(4)*	50 *(4)*	50 *(4)*	25 *(4)*	33 *(3)*	25 *(4)*	- *(0)*
*Phaeoacremonium* spp.	2/3	50 *(2)*	50 *(2)*	50 *(2)*	100 *(2)*	0 *(1)*	100 *(2)*	- *(0)*
*Scedosporium* spp.	12/16	100 *(12)*	83 *(12)*	0 *(12)*	33 ^c^ *(12)*	75 *(4)*	50 *(12)*	100 *(2)*
*Scopulariopsis* spp.	1/3	100 *(1)*	100 *(1)*	100 *(1)*	100 *(1)*	- *(0)*	0 ^d^ *(1)*	- *(0)*
*Trichoderma* spp.	4/4	25 *(4)*	75 *(4)*	0 *(4)*	50 *(4)*	75 *(4)*	0 *(4)*	- *(0)*

AmB: amphotericin B; ITZ: itraconazole; VRZ: voriconazole; PSZ: posaconazole; ISZ: isavuconazole; CAS: caspofungin; 5FC: 5-flucytosine. ^a^ Among *Fusarium* spp., one had a PSZ MIC of 1 mg/L, and one had a PSZ MIC of 2 mg/L; both were considered resistant. ^b^ Among *Paecylomyces* spp., one had a PSZ MIC of 0.75 mg/L and was considered resistant. ^c^ Among *Scedosporium* sp., one had a PSZ MIC of 0.25 mg/L and was considered susceptible, two had a PSZ MIC of respectively 0.38 and 0.5 mg/L and were considered of intermediate susceptibility and five had a PSZ MIC between 1 and 2 mg/L and were considered resistant. ^d^ *Scopulariopsis* sp. had a CAS MIC of 2 mg/L and was considered of intermediate susceptibility. There were no available data on MIC for *A. kalrae*, *C. bantiana*, *Cladosporium* sp., *L. prolificans*, *P. romeroi*, or *Schizophyllum* sp. Cells are colored green when less than 25% of species tested exhibited MIC > 2 mg/L, yellow when percentage of species with MIC > 2 mg/L is comprised between 25 and 74% and red when 75% or more of the species tested exhibited MIC > 2 mg/L.

## Data Availability

The data presented in this study are only available on request from the corresponding author due to privacy regulations.

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
