# Peer review of "Fusarium, Scedosporium and Other Rare Mold Invasive Infections: Over Twenty-Five-Year Experience of a European Tertiary-Care Center"

_jof, 2024, doi:10.3390/jof10040289_

Round 1

Reviewer 1 Report

This retrospective study provides valuable insights into infections caused by this group of fungi over an extended period. A key critique would be to delve deeper into the methodology employed for species-level identification. Further comments and corrections are included in the pdf.

The information presented in Table 1 is somewhat challenging to follow. It contains an abundance of details presented in a continuous manner.

Author Response

Dear reviewer 1, 

We are deeply grateful for your attentive reading of our manuscript and your relevant suggestions. 

As an answer to your review, we amended our article by incorporating writing corrections of style and form, as well as some more details about fungi identification technics. 

About the title issue, we eventually decided to stand by it. Of course we agree that Fusarium and Scedosporium are not infections, but they are molds, and "Invasive infections" can be distributed to the different epithets of the title: Fusarium, Scedosporium and other molds. We think the reader will be more acquainted with the word Fusarium than with the word fusarisosis, and with the word Scedosporium than with the word scedosporiosis. Moreover, in the alternative title "Fusariosis, scedosporiosis and other rare molds invasive infections", it is not obvious that we are talking about invasive fusariosis and invasive scedosporiosis only.  

About the Scedosporium and Lomentospora issue, your comment decided us into more clearly separating the categories. We recalculated the percentages implied, with in the end little difference due to the fact that the only infection involving a Lomentospora in our cohort was a co-infection also involving a Scedosporium sp.

As you will see in the revised manuscript, we also added data about antifungal susceptibility and trends over time, following the suggestions of the other reviewers. We hope the additionnal material will be of some interest for you. 

Once again, we sincerely thank you for your relevant and useful review. 

Reviewer 2 Report

Is aplastic anaemia really a benign haematological disorder?

Table 1 - age should IQR not range. Similarly with duration of neutropenia

Table 2 and the comments areound it should be in the methods section and not in the results. The actual data around this should be in the results section.  

Line 118 - no need to say it is a pier chart. Just say shown in Figures 1.

Line 187 - need to add the IQR as well

Should add in data and a figure of trends over time. It is 25 years you are looking at.   

Line 309 - should be number of blood cultures and not number of tests.

In the results section, you should differentiate between Scedosporium spp. and Lomentospora more. I know that Lomentospora was found only once and it was part of  mixed infection. I think that given they are now differentiated and we are really wanting to see separate data on these (Scedosporium spp .and L. prolificans) it would be good to add a few sentences on this in the results. It is discussed in the discussion but we should see some more in the results section.   

No discussion about the time-span of the study and how the diagnosis, management and treatment can change over time. So many things have changed over time that may affect the outcomes over time.  This should be looked at in the results section and discussed.  

A number of grammatical/English issues that need to be changed altered

Line 19 - bearers. maybe patients

Line 30 - awaited. Maybe needed

Line 39 - of should be at

Lines 46-47 - need to change altogether. 

Line 49 - frames. Maybe frontiers

Line 50 - raise. Maybe increase 

Line 101 - of should be with 

Line 106 - should be near drowning 

Line 121-122 - great idae. But need to make it more explicitly. Something like Tabel 2 outlines the old and new names for fungi detected in this study. 

Line 166 - positive in is better

Line 217-218 - just use median and IQR, no need for mean and range as well

Line 256 - to diverse background patients. Needs to be changed. Maybe - patients with different underlying diseases

Line 260 - should be in our cohort

Line 273 - should be than and not that 

LINE 275 -Transnet should have all letters capitalised

Line 332 - should be outcomes

Line 339 - should be ranking of 

Line 343 - Take out anyway. This is not a scholarly word

Author Response

Dear reviewer 2, 

We are deeply grateful for your attentive reading of our manuscript and your relevant suggestions. 

As an answer to your review, we amended our article by incorporating writing corrections of style and form. Along your suggestion, we reorganised former table 2 (now 1) and comments around it. We added IQR wherever you suggested it, and also kept the ranges which we felt might give an additionnal comprehension of the cohort. 

About the Scedosporium and Lomentospora issue, your comment decided us into more clearly separating the categories. We recalculated the percentages implied, with in the end little difference due to the fact that the only infection involving a Lomentospora in our cohort was a co-infection also involving a Scedosporium sp.

About aplastic anemia, it can be by all means severe, but it is still non malignant, and that's why we make it fall into the "benign hematological disorders" category, not unlike other authors. 

We tried our best to provide the reader with some additionnal data that could point towards a trend in time. As you will see, our cohort does not enable us to conclude on a clear improvement. We hope our answer to your suggestion will appear relevant. 

As you will see in the revised manuscript, we also added data about antifungal susceptibility, following the suggestions of the other reviewers. We hope the additionnal material will be of some interest for you. 

Once again, we sincerely thank you for your relevant and useful review. 

Reviewer 3 Report

Few studies report co-infections caused by rare molds. The cohort covers a long period of 25 years.

This work highlights the importance of accurate diagnosis of fungal infection. It is important for clinicians to keep in mind that when an Aspergillus infection is not responding to treatment, they may address the possibility of co-infection with another fungus that may be resistant to the antifungal treatment adopted.

The images illustrating 2 clinical cases are interesting as examples for readers.

Evaluation of biomarkers tests for detecting different mold species is also of interest for clinicians, this information is rarely available in the literature.

On section Patients and Method, page 2 line 82, the authors report that MICs were determined with a gradient method. They did not specify the methodology used neither report the MICs values for the isolates studied. This information is missing on the manuscript and is of great value for the medical community that read this paper, so I recommend that these data are included in the manuscript.

Discussion is well conducted but should include comments on the resistance detected to the antifungal tested. This issue will be of great interest to the medical field.

Conclusions are proper to the presented data.

On section Patients and Method, page 2 line 82, the authors report that MICs were determined with a gradient method.

Author Response

Dear reviewer 3, 

We are deeply grateful for your attentive reading of our manuscript and your relevant suggestions. 

Following your suggestion on antifungal susceptibility, we went back to the database to extract all the available MIC in our cohort and tried to present them in a way that could be useful and reader-friendly. We hope that you will find our answer relevant and wish to thank you for pointing out this neglected part of our study. 

As you will see in the revised manuscript, we also added data about trends over time, following the suggestions of the other reviewers. We hope the additionnal material will be of some interest for you. 

Once again, we sincerely thank you for your relevant and useful review. 

Round 2

Reviewer 1 Report

The content of the article is important as it refers to the results obtained on fungal infections over a long period. Probably for this reason, the section corresponding to the techniques used for the identification of the different strains is the least developed and specified, but this is understandable considering the large volume of information covered.

The content of the article is important as it refers to the results obtained on fungal infections over a long period. Probably for this reason, the section corresponding to the techniques used for the identification of the different strains is the least developed and specified, but this is understandable considering the large volume of information covered.